# Prolonged B-Lymphocyte-Mediated Immune and Inflammatory Responses to Tuberculosis Infection in the Lungs of TB-Resistant Mice

**DOI:** 10.3390/ijms24021140

**Published:** 2023-01-06

**Authors:** Irina Linge, Elena Kondratieva, Alexander Apt

**Affiliations:** Laboratory for Immunogenetics, Central Tuberculosis Research Institute, Moscow 107564, Russia

**Keywords:** tuberculosis, B cells, chronic infection, IL-6, neutrophils, metalloproteinases

## Abstract

During tuberculosis (TB) infection, B-lymphocytes migrate to the lungs and form B-cell follicles (BCFs) in the vicinity of TB granulomata. B-cell-lacking mice display enhanced susceptibility to TB infection, and early B-cell depletion in infected non-human primates alters T-lymphocyte cytokine responses and increases bacterial burdens in the lungs. However, the role of B cells during late TB stages remained unaddressed. Here, we demonstrate that B cells and BCFs persist up to weeks 25–45 post-challenge in the lungs of TB-resistant C57BL/6 (B6) mice. In hyper-susceptible I/St mice, B-cell content markedly drops between weeks 12–16 post-infection, paralleled by diffuse lung tissue inflammation and elevated gene expression levels for pro-inflammatory cytokines IL-1, IL-11, IL-17a, and TNF-α. To check whether B-cells/BCFs control TB infection at advanced stages, we specifically depleted B-cells from B6 mice by administrating anti-CD20 mAbs at week 16 post-infection. This resulted in more rapid cachexia, a shortened lifespan of the infected animals, an increase in (i) lung-infiltrating CD8^+^ T cells, (ii) IL-6 production by F4/80^+^ macrophages, (iii) expression levels of genes for neutrophil-attracting factors CXCL1 and IL-17, and tissue-damaging factors MMP8, MMP9, and S100A8. Taken together, our results suggest that lung B cells and BCFs are moderately protective against chronic TB infection.

## 1. Introduction

Tuberculosis (TB) remains a global health problem, resulting in over 1 million deaths annually [1]. It is commonly recognized that macrophages and CD4^+^ T cells are the main players in immune responses against *Mycobacterium tuberculosis* (Mtb), the causative TB agent [2,3]. However, other immune cell types are involved in immune and inflammatory reactions to mycobacteria. Thus, it was firmly established in humans [4,5] and animal models [6,7,8,9,10] that during the course of TB, B cells migrate to lung tissue and form B-cell follicles (BCFs) in close vicinity of granulomata. Lung BCFs possess all prominent features of B-cell follicles of secondary lymphoid organs, including the presence of Tfh CXCR5^+^ cells, follicular DC, high endothelial venules, and the formation of germinal centers [8,10,11]. Despite rising interest in B-cell physiology in TB, the exact role of lung B cells in anti-TB response is still not sufficiently determined. TB challenge of mice with genetically abrogated B-cell populations demonstrated increased lung neutrophil influx, impaired formation of lung TB granuloma, and a decreased life span post-challenge [12]. Depletion of B cells with Rituximab prior to and during early phases of TB infection in non-human primates (NHP) resulted in decreased IL-6 and IL-10 production, increased frequencies of IL-2-, IL-10-, and IL-17-producing CD4^+^ T cells, and an increased CFU count in individual lung granuloma of treated animals [11]. These results suggest a protective role of B cells in the acute phase of infection. Our recent study [13] also demonstrated that lung B lymphocytes play a defensive role at the early stage of TB infection due to a high level of IL-6 secretion by these cells. Meanwhile, the role of B cells and lung BCFs during advanced stages of TB infection remains unaddressed, although certain cell populations may play different roles at the initial and advanced stages of chronic diseases. Thus, it was shown that exactly at the advanced phase of chronic obstructive pulmonary disease (COPD), the numbers of lung-infiltrating B cells and BCFs are associated with emphysema development and more severe pathology [14], whilst during acute COPD, the exacerbation of B cells plays a mechanistically non-explained but protective role [15].

Animal TB models present a wide variety of phenotypes reflecting TB infection development. Thus, in cynomolgus macaques, TB infection causes all types of human-like lung pathology [16]. Although the relevance of mouse TB models is a constant subject of debate, various genetically TB-susceptible mouse strains also develop human-like TB, which allows adequate investigation of different aspects of TB inflammation. Some inbred mice, e.g., C3HeB/FeJ, DBA/2, CBA/J, and I/St as well as specifically selected outbred stocks develop human-like necrotic granulomata surrounded by hypoxic tissue during TB infection [9,17]. Meanwhile, C57BL/6 (B6) mice most commonly used in TB research are relatively resistant to infection and do not develop lung necrosis [9]. Instead, these mice may serve as a model appropriate for studying slow-progressing, chronic TB and resistance-providing factors. Comparisons performed in different mouse TB models revealed the following characteristic features of B-cell activity during the course of TB: First, B cells form lung BCFs both in resistant and susceptible mice; however, the dynamics of this response differ between strains. After a low-dose aerosol infection of susceptible I/St mice, B cells accumulate in the lungs between weeks 3 to 8 post-infection followed by peaking of B-cell and BCF numbers at weeks 8–10 post-infection. Subsequently, the numbers of BCFs and B-cells themselves diminish [10], whereas in resistant B6 mice, lung BCFs persist much longer. Second, a comparison between B6 males and females after the aerosol TB challenge demonstrated that a higher level of susceptibility in males is associated with a smaller number of lung BCFs at the late stage of infection [18]. Third, after infection with clinical HN878 Mtb isolates, genetically susceptible C3HeB/FeJ mice formed fewer lung BCFs compared to resistant B6 mice [19]. Collectively, these results suggest a protective role of B cells in the local lung-associated TB immune response, but mostly the evidence remains indirect.

To further clarify this issue, here, we compared the dynamic of B-cell accumulation and BCF formation in our model of low-dose aerosol *M. tuberculosis* H37Rv infection of TB-susceptible I/St and relatively resistant B6 mice. We demonstrate that in I/St mice, BCFs and B cells disappear from the lung tissue around weeks 12–16 post-infection, whereas a significant number of B cells and BCFs persist up to weeks 25–45 in the lungs of resistant B6 mice. Vanishing of B cells from the lungs of I/St mice coincides with the progression of severe lung pathology, diffuse inflammation, formation of necrotic zones, and significant elevation in the expression levels of genes for pro-inflammatory cytokines IL-1, IL-11, IL-17a, and TNF-α compared to B6 mice. Anticipating that B cells/BCFs participate in the control of TB inflammation during the chronic stage of the infection in resistant B6 mice, we checked this hypothesis by specifically depleting B cells from B6 mice with anti-CD20 mAbs at week 16 post-infection. Administration of highly effective depleting antibodies led to more severe cachexia and a decreased lifespan of the infected animals. B-cell depletion only moderately affected T-cell recruitment to the lungs, resulting in an increase in CD8^+^ but not CD4^+^ lung T-cell population. Additional assessments of IL-6 production and expression of genes for neutrophil-associated factors allow for the conclusion that lung B cells/BCFs have moderate but clear protective effects in TB-infected lungs.

## 2. Results

### 2.1. Early Reduction in BCFs/B-Cell Numbers in TB-Susceptible I/St Mice Coincides with Lung Pathology and Inflammation Progression

We previously showed that B6 and I/St mice differ in their susceptibility to tuberculosis [9,20] and investigated the dynamic of B-cell migration to the lungs of I/St mice during a TB course [10]. In the present work, we first compared B-cell migration and BCF formation in I/St and B6 mice. In susceptible I/St mice, the numbers of BCFs and B cells in the lung tissue remarkably dropped by week 16 post-infection, whereas a significant number of B cells and BCFs persisted at week 20 post-challenge in the lungs of B6 mice (Figure 1A–C); moreover, their presence was still observed as late as week 45 post-infection (Appendix A). Remarkably, diminishing the numbers of BCFs in the lungs of I/St mice (Figure 1C, upper panels) coincided with the development of diffuse lung inflammation, while their persistence in B6 lungs was accompanied by the presence of structured granulomatous foci delineated from surrounding tissue (Figure 1C, lower panels). At the late stage of infection, B cells were readily observed both within BCFs and in the lung parenchyma of the B6 mice (Appendix A). At week 12 post-infection, we also assessed the expression levels of genes for a few key inflammatory cytokines. In the lung tissues of I/St mice, significantly elevated levels of *il1*, *il11*, *il17a*, and *tnfa* RNA encoding, respectively, IL-1, IL-11, IL-17, and TNF-α were observed compared to B6 mice (Figure 1D). These results suggest that long-term maintenance of BCFs in the lungs is associated with better control of tuberculous lung inflammation.

### 2.2. In Vivo Depletion of B Cells during Advanced TB Infection Is Detrimental

We next evaluated the putatively defensive role of B cells in TB at the advanced stage of the infection with a widely used experimental approach—in vivo depletion of the population of cells in question by administration of specific cytotoxic antibodies. To this end, we IV injected TB-infected B6 mice with anti-CD20 B-cell-specific antibodies. Although the manufacturers of these antibodies claimed that after a single injection, B cells disappear from circulation for about 30 days, in our TB model, restoration of B cells in the peripheral blood started at day 7–10 post-treatment; thus, we injected antibodies thrice starting at week 16 of infection with 10-d intervals (Figure 2A). After a single antibody injection, we observed an almost complete disappearance of B cells from the blood and a significant shrinking of lung BCFs (Figure 2B), the phenotype supported by subsequent antibody injections for at least 30 days (Figure 2C). We then evaluated the influence of B-cell depletion on key parameters of TB progression, that is, the mycobacterial burden in the organs, cachexia, and lifespan of the infected animals. Evaluation of the CFU counts in the lungs and spleens during the phase of B-cell absence demonstrated only marginal to no differences between the groups of mice (Figure 2F). However, comparison of late-disease phenotypes, such as cachexia progression and survival time, revealed significant differences between the groups: mice depleted of B cells at weeks 16–20 of the infectious course lose body weight more rapidly after week 45 and survived for a shorter period than the control animals (Figure 2D,E).

### 2.3. The Influence of B Cell Depletion on T Cell Response

During TB infection, the majority of lung B cells are located within the BCFs where they are tightly in contact with CD4^+^ T cells ([10,21] and Appendix A); the latter are generally considered one of the main defensive components in anti-TB response. To assess the possible impact of B-cell elimination on the T-cell branch of anti-TB immunity in the lungs, we measured the size of two major T-cell populations and the inflammatory cytokine responses in B-cell-depleted and control animals. To our surprise, B-cell depletion influenced neither the frequency of lung CD4^+^ T cells (Figure 3A) nor their activation status (Figure 3B). We also failed to detect differences in the number of lung CD4^+^ T cells producing IFN-γ, TNF-α, or IL-17 in response to mycobacterial cultural filtrate stimulation in the B-cell-depleted and control mice (Figure 3C). A slight increase was observed for the frequency (but not the total number per organ) of lung CD8^+^ T cells in the B-cell-depleted mice (Figure 3D); however, their activation level was similar to that of control animals (Figure 3E). Contrary to the CD4^+^ T cells, decreased numbers of TNF-α- and IL-17-producing mycobacteria-specific CD8^+^ T cells in the absence of B cells were detected (Figure 3F).

### 2.4. B Cell Depletion Increases IL-6 Production by Lung Macrophages

Previously, it was shown that B-cell depletion during early phases of TB infection leads to a decreased IL-6 production in the lungs [11] and that B cells are an important source of IL-6 in early anti-TB response [13]. In this study, we measured in vitro IL-6 production by lung cells from B-cell-depleted and control B6 mice after their stimulation with mycobacterial cultural filtrate. Unexpectedly, at the advanced stage of TB infection, B-cell deficiency resulted in an increased IL-6 production by lung cells (Figure 4A). Macrophages are generally considered a major source of IL-6 during chronic inflammatory processes [22,23], including pulmonary TB [24]. When we double-stained lung cell suspensions for immune cell markers and intracellular IL-6, it appeared that the proportion of CD11b^+/med^IL-6^+^ phagocytes was higher in B-cell-depleted compared to control mice (Figure 4B right panel). At the same time, Ly-6G^+^ (CD11b^hi^) neutrophils were IL-6-negative, and B cells and CD4^+^ T cells were less than 2 percent of all IL-6-producing cells whereas the vast majority of IL-6-producing cells (more than 60%) were represented by F4/80^+^ macrophages (Figure 4C). Interestingly, neither the frequency nor the total number of F4/80^+^ macrophages differed between B-cell-depleted and control mice (Figure 4D), suggesting that the difference displayed in Figure 4B,C reflects exactly a shift towards IL-6 synthesis in a larger proportion of the cells. Since one of the triggers of IL-6 synthesis in macrophages is an accumulation of cholesterol and other fats in vacuoles [25], we stained the lung tissue of mice with the OilRedO reagent to visualize the fat content. As shown in Figure 4E, the lungs of B-cell-depleted mice indeed contained much more positively stained cells both in the zones of diffuse inflammation and in delineated TB foci. Thus, in the absence of B cells in the lungs at the advanced stage of TB infection, classical F4/80^+^ macrophages are the major source of IL-6.

### 2.5. B Cell Depletion Increases the Content of Neutrophil-Associated Factors of Lung Pathology

It is well-established that on the background of B-cell deficiency, a detrimental increase in neutrophil recruitment to the sites of mycobacteria location occurs [12,26,27]. In the model described herein, we observed only a non-significant increase in the numbers of lung-infiltrating neutrophils in B-cell-depleted and control mice (Figure 5A). However, at the mRNA level expression of both genes for neutrophil recruiting factors CXCL1 and IL-17 (Figure 5B), and genes for neutrophil-associated inflammatory and tissue-damaging factors S100A8 and matrix metalloproteinases MMP8 and MMP9 (Figure 5C) were significantly elevated in the absence of B cells. Thus, B-cell deficiency creates a transcriptional milieu favorable for neutrophil-associated inflammation, which is detrimental to TB containment.

As opposed to humans, fibrotic metamorphosis is not considered a common TB feature in murine experimental TB. Nevertheless, in I/St mice hyper-susceptible to infection, we clearly observed fibrotic processes (SMA-positive myofibroblasts infiltration) at week 9 post-infection (Figure 5D), prompting us to check whether this type of TB pathology depends upon B-cell presence. Whereas practically no fibrotic zones were present in the lungs of the control mice, the arrival of SMA-positive myofibroblasts was evident in the lungs of the B-cell-depleted animals (Figure 5E), demonstrating for the first time that B cells play a role in this aspect of TB pathology.

## 3. Discussion

B cells participate in immune responses to many intracellular bacteria, including *Legionella pneumophila* [28,29], *Coxiella burnetii* [30], *Brucella abortus* [31,32,33], *Chlamydia trachomatis* [34,35], *Francisella tularensis* [36], *Leishmania major* [37], *Plasmodium chabaudi chabaudi* [38], *Pneumocystis carinii* [39], and *Salmonella entericasero* var *Typhimurium* [40]. In regard to B cell involvement in anti-mycobacterial responses, according to the latest data, differently glycosylated specific antibodies detected in serum distinguish individuals with active TB, latent TB, and healthy contacts [41,42,43]. There is still no direct proof regarding lung B-cell impact on serum antibody levels, although it was shown that B cells from the lungs of infected animals produce mycobacteria-specific antibodies [10,11].

Besides antibody production, B cells in TB-infected lungs produce cytokines, serve as antigen-presenting cells (APC) and interact with neighboring immune cells [7,8,10,11,44,45]. Experiments in B-cell-deficient mouse strains [6,12,46,47,48] resulted in rather controversial conclusions about the influence of B cells on TB progression. It is worth mentioning that B-cell KO mouse models are often a subject of debate since such mice have an altered structure of all immune organs, potentially influencing anti-TB responses. Application of an alternative approach, that is, B-cell depletion in vivo with specific cytotoxic antibodies, shed some light on the role of B cells in the NHP TB model during the acute phase (up to 10 weeks post-challenge) of infection and provides new knowledge about B-cell performance within individual TB granuloma [11]. In that study, further monitoring was abandoned since the animals stopped responding to Rituximab treatment; thus, the role of B cells at the chronic TB stage remained unaddressed. However, evidence of a decrease in the number and sizes of BCFs in the lungs along TB progression was obtained in mouse TB models [18,19].

Here, we compared mouse strains with polar TB susceptibility and found out that in TB-susceptible I/St mice, lung B-cell population and BCF numbers gradually diminish starting week 8 post-challenge, which approximately coincided with the development of diffuse lung tissue inflammation (Figure 1). However, TB-resistant B6 mice retained lung BCFs for a much longer time, which provided an opportunity to study possible roles of B cells and BCFs during chronic TB stages. B-cell depletion with anti-CD20 antibodies at weeks 16–20 post-challenge demonstrated only marginal to no influence on the mycobacterial burden in organs but enhanced cachexia progression and shortened the life span of the animals (Figure 2).

In TB-infected B6 mice, B cells are located predominantly within the BCFs, although their population is partly scattered in the lung parenchyma (Appendix A). In both TB patients and TB-infected animals, it was shown that besides B cells, the vast majority of immune cells within BCFs are CD4^+^ and CD8^+^ T lymphocytes [4,8,10] (Appendix A). Since lung granuloma may serve as an efficient structure for T-cell priming, even in animals without secondary lymphoid organs [49], we evaluated the possible influence of B-cell depletion on CD4^+^ and CD8^+^ T-cell populations. Surprisingly, B-cell deficiency had no effect on the size of the CD4^+^ T-lymphocyte population or mycobacteria-specific inflammatory cytokine production (Figure 3). On the contrary, the CD8^+^ T-cell population size was elevated in B-cell-deficient mice compared to controls, which, unexpectedly, was accompanied by a decreased frequency of cells producing TNF-α and IL-17 after stimulation with mycobacterial antigens. These results are in agreement, at least in part, with observations in a B-cell −/− low-dose aerosol mouse TB model in which B-cell-deficient mice displayed unaltered CD4^+^ T-cell response but an increased lung infiltration with CD8^+^ T cells [12].

The number of experimental results concerning various outcomes of interactions between T- and B-lymphocytes in TB infection models is impressive; however, we are still far from understanding not only the exact immune mechanisms but also the precise dynamics of particular responses. While in a few studies, no difference in CD4+ T-cell responses, including IFN-γ phenotypes, were observed on the background of Ab- [50] or B-cell deficiency ([12,47] 2007, and this work), some works substantiate different conclusions. Thus, NHP treated with Rituximab for B-cell depletion prior to and during the early phases of TB infection displayed elevated frequencies of T cells producing IL-2, IL-10, and IL-17 [11]. We recently demonstrated that IL-6 deficiency in B cells leads to a delayed mycobacteria-specific T-cell response [13]. To our opinion, these discrepancies may reflect the different values of the APC B-cell function at the early and advanced stages of TB infection. After classical initiation of T-cell response by mycobacteria-contacted dendritic cells during weeks 2–2.5 of infection, at least two additional cell types with APC capacities rapidly accumulate in the lung tissue: BCF-forming B cells dealing with soluble mycobacterial antigens and lung macrophages containing engulfed mycobacteria. The latter cells produce IL-12 and effectively activate T cells [2]. It is fairly possible that on the background of fully established T-cell immunity, interaction between B-cells and CD4^+^ T-cells is less critical and its local maintenance drops.

During the course of TB, a significant proportion of lung macrophages transforms into fat-rich, foamy macrophages, the hallmark of lung TB granuloma [51] and a major contributor to TB pathogenesis [52,53]. We observed a marked enrichment in fat-positive cells in B-cell-deficient mice (Figure 4E). Interestingly, in a non-TB setting, cholesterol accumulation in human macrophages resulted in elevated production of IL-6, which in turn protected macrophages from further cholesterol accumulation [25]. We detected an elevated level of overall IL-6 production (Figure 4A) and observed an increased number of IL-6-producing macrophages (Figure 4B–D) in the B-cell deficient environment. An increased IL-6 production might be a counter-reaction to the accumulation of lipids, and the question of whether B cells can directly influence the formation of foamy macrophages deserves further investigation.

Finally, B-cell depletion at the chronic stage of TB infection resulted in an increased expression in the lung tissue of genes for neutrophil-associated responses: neutrophil-attracting factors CXCL1 and IL-17, tissue-damaging MMP8 and MMP9, and inflammatory S100A8 (Figure 5B,C). S100A8/A9 dimer protein is produced by neutrophils and mediates additional neutrophil attraction, inflammation, and lung pathology during TB [54,55]. S100A9-negative KO mice demonstrated improved control of chronic (300 days post-challenge) TB infection caused by clinical *M. tuberculosis* isolates HN878 and HN563. Moreover, enhanced TB resistance of S100A9-KO mice was associated with higher numbers of lung BCFs at the chronic stage of infection [55]. Neutrophil-associated metalloproteinases MMP8 and MMP9 both mediate matrix destruction during pulmonary TB and contribute to subsequent fibrosis formation [56]. In our experiments, only B-cell-depleted mice developed fibrosis, albeit at a moderate level. Moreover, fibrotic areas were practically B-cell-free, whereas no SMA-positive cells were present near B-cell follicles (Figure 5E), which suggests mutual negative bilateral cell interaction.

Overall, our results suggest that even if TB infection developed in immune-competent animals and immune responses were established under normal conditions, B-cell deficiency at an advanced stage of infection exacerbates the disease. This is featured by a more rapid accumulation of factors worsening lung pathology and ends up with more severe cachexia and earlier mortality of infected animals. Taken together with B-cell response characteristics obtained previously at the early stages of TB infections, we may conclude that B-cell influx in TB-affected lungs plays a moderately protective rather than pathogenic role during the course of TB.

## 4. Materials and Methods

### 4.1. Mice

Mice of the C57BL/6JCit (B6) and I/StSnEgYCit (I/St) strains were bred under conventional, non-SPF conditions at the Animal Facilities of the Central Institute for Tuberculosis (Moscow, Russia) in accordance with the guidelines from the Russian Ministry of Health # 755, US Office of Laboratory Animal Welfare (OLAW) Assurance #A5502-11. Water and food were provided *ad libium*. Female mice 10–12 weeks of age at the beginning of experiments were used. All experimental procedures were approved by the Institutional Animal Care and Use Committee (IACUC), protocols 1, 2, 3, and 10 of 2 March 2021.

### 4.2. Infection, CFU Counts, and Survival Time

Mice were infected with 10^2^ CFU of virulent *M. tuberculosis* strain H37Rv (sub-strain Pasteur) using the inhalation Exposure System (Glas-Col, Terre Haute, IN, USA) exactly as previously described [57]. At different time points post-challenge (as indicated in Figure 1 and Figure 2), mice were euthanized by thiopental (Biochemie GmbH, Vienna, Austria) overdose. Spleens were homogenized in 2.0 mL of sterile saline and middle lobes of the right lung from individual mice were digested in collagenase-DNase 1 mixture as described below except for the presence of antibiotics. Then, 10-fold serial dilutions of 0.2 mL samples were plated on Dubos agar (Difco, Sparks, MD, USA) and incubated at 37 °C for 20–22 days before CFU were counted. Survival time and weight loss were monitored starting month 3 post-infection.

### 4.3. Lung Cell Suspensions

Middle right lobes of lungs from individual mice were enzymatically digested exactly as described previously [58]. Basal right lobes were used for RNA isolation and purification. Briefly, blood vessels were washed out by perfusion with 0.02% EDTA-PBS through the right ventricle and cut vena cava; lung lobes were removed, sliced into 1–2 mm^3^ pieces, and incubated at 37 °C for 90 min in supplemented RPMI-1640 containing 200 U/mL collagenase and 50 U/mL DNase-I (Sigma, St. Louis, MO, USA). Single-cell suspensions from 4–5 mice were obtained individually and washed twice in HBSS containing 2% FCS and antibiotics. Next, 3 × 10^5^ cells from each sample were used for surface phenotyping, whereas 1 × 10^6^ cells were cultured for 18 h in the absence (control) or presence of 10 μg/mL mycobacterial antigens mixture and the last 12 h of incubation with GolgiPlug (BD Bioscience, San Jose, CA, USA) for further evaluation of intracellular cytokines. Antigenic mixture was mycobacterial cultural filtrate (CF) (kindly provided by Dr. V. Avdienko, CTRI). Briefly, CF was obtained by culturing *M. tuberculosis* as a biofilm in protein-free Sauton’s medium for 4 weeks followed by precipitation of mycobacteria at 3000× *g* for 20 min, concentration of the protein content in the liquid phase using Amicon cell device, and bringing the final protein concentration to 1 mg/mL with sterile saline.

### 4.4. B Cell Depletion

Purified anti-mouse CD20 antibodies (SA271G2, Ultra-LEAF™, BioLegend, Heidelberg, Germany) were injected intravenously, 0.25 mg/mouse at week 16 (point 0, p0) and then twice with 10 d intervals.

### 4.5. Flow Cytometry

Single-cell suspensions were obtained from individual mice, and cell phenotypes were analyzed by flow cytometry using FACS Canto II (BD Biosciences, San Jose, CA, USA). The following labeled monoclonal antibodies were used: CD4-BV421 (GK1.5, BioLegend, Heidelberg, Germany), CD8+ (53–5.8, BioLegend, Heidelberg, Germany), CD19-BV510 (6D5, BioLegend, Heidelberg, Germany), F4/80-AF-488 (BM8, BioLegend, Heidelberg, Germany), Ly6G-PE (1A8, BD Biosciences, San Jose, CA, USA), CD44-FITC (IM7, BioLegend, Heidelberg, Germany), CD62L-PE (MEL-14, BioLegend, Heidelberg, Germany), CD11b-biotin (M1/70, BioLegend, Heidelberg, Germany) with subsequent SAv-PerCP (dilution 1:500, BioLegend, Heidelberg, Germany) staining. For intracellular staining of the lung cells, BD Fixation/Permeablization Kit (BD Biosciences, San Jose, CA, USA) was used as recommended by the manufacturer. Briefly, after 18 h of in vitro cultivation without (control; Appendix A) or in the presence of mycobacterial antigens (last 12 h also in the presence of BDGolgyPlug), cells were harvested, stained for surface antigens, fixed with Cytofix/cytoperm buffer for 30 min, and stained for intracellular cytokines with IFN-γ-APC (dilution 1:150, BD Biosciences, San Jose, CA, USA), IL-17-PerCp (TC11-18H10.1, BioLegend, Heidelberg, Germany), TNF-α-FITC (BioLegend, Heidelberg, Germany), and IL-6-PE (BioLegend, Heidelberg, Germany) antibodies. Results are displayed as mean ± SD. Gating strategies for every estimated cell population are shown in Appendix A.

### 4.6. Histology and Immunohistochemistry (IHC)

Left lungs were frozen at the regimen of −20 °C to −60 °C temperature gradient in the electronic cryotome (ThermoShandon, Oxford, UK), and serial 10 μm thick sections were made across the widest area. For visualization of lung pathology, sections were fixed with ice-cold acetone and stained with hematoxylin and eosin. For IHC, lung cryosections were fixed with 1% PFA, blocked with 10% donkey serum, and incubated with rat anti-mouse primary B220 (IgG2a, clone RA3-6B2, eBioscience, San Diego, CA, USA), rabbit anti-mouse primary anti-CD3 (clone SP7, ThermoScientific, Waltham, MA, USA), SMA-Cy3 (clone 1A4, Sigma, St. Louis, MO, USA), ERTR7-AF488 for 1 h at room temperature. Afterward, preparations were incubated with secondary donkey anti-rabbit IgG-Cy3, donkey anti-rat IgG-AF488, or anti-rabbit IgG-AF488 pAB for 1 h at room temperature. Slides were preserved using ProLong Gold anti-fade reagent with DAPI (Invitrogen-Life Technologies, Waltham, MA, USA) before visualization using the Zeiss Axioskop40 microscope and AxioCam MRc5 AxioVisio 4 camera (Carl Zeiss, Berlin, Germany).

To visualize lipid droplets in macrophages, cryosections on SuperFrost Plus glasses were air-dried, placed directly into working solution of filtered 0.5% Oil red O (Sigma-Aldrich, St. Louis, MO, USA) in absolute isopropyl, and mixed with 1% Dextrin (Sigma-Aldrich, St. Louis, MO, USA) in distilled water for 20 min. Slides were briefly rinsed with tap water, counterstained with hematoxylin (Thermo Shandon, Cheshire, UK) for 20–30 s, rinsed again with tap water, and mounted with aqueous media (Thermo Shandon, Cheshire, UK).

### 4.7. Cytokine ELISA

1 × 10^6^/mL lung cells were cultured in wells of 24-well plates for 48 h in the presence of 10 μg/mL mycobacterial ultrasonic disintegrate, established as previously described [58]. IL-6 cytokine contents in supernatants were assessed using DuoSet ELISA kit (R&D systems, Minneapolis, MN, USA) according to the manufacturer’s instructions.

### 4.8. RNA Purification, cDNA Synthesis, and Gene Expression Evaluation

Identical lobes of the right lungs from 4–5 individual mice per group per time point were isolated and immediately put into RNA lysis buffer. Total RNA was isolated using the SV Total RNA Isolation System (Promega, Madison, WI, USA) according to the manufacturer’s recommendations. Reverse transcription of mRNA was performed using oligo(dT) primers, dNTP mix, M-MLV RT, and RNasin^®^ (Promega, WI, USA). Quantitative real-time PCR (qrt-PCR) was performed using qPCRmix-HS SYBR (Evrogen, Moscow, Russia) and cfx-96 Real-Time PCR Detection System (BioRad, Hercules, CA, USA). The following primers were used:

*hprt*:

F 5′-GTGATTAGCGATGATGAACCAG-3′,

R 5′-CAAGTCTTTCAGTCCTGTCCA-3′;

*il6*:

F 5′-TCTATACCACTTCACAAGTCG-3′,

R 5′-TAGGCAAATTTCCTGATTATATCCA-3′;

*il11*:

F 5′-ACATGAACTGTGTTTGTCGC-3′,

R 5′-ATCGGGTTAGGAGAACAGC-3′;

*il17a*:

F 5′-CCAGAATGTGAAGGTCAACC-3′,

R 5′-TTCATTGCGGTGGAGAGTC-3′;


*tnfa:*


F 5′-TTCTGTCTACTGAACTTCGGG-3′,

R 5′-AGAAGATGATCTGAGTGTGAGG-3′;

*cxcl1*:

F 5′-AACCGAAGTCATAGCCACAC-3′,

R 5′-AGAGCAGTCTGTCTTCTTTCTC-3′;


*mmp8:*


F 5′-CTCTCACTCCACTGATCCTG-3′,

R 5′-TGTCTGAAGGTCCATAGATTGTC-3′;


*mmp9:*


F 5′-CTTGAAGTCTCAGAAGGTGGA-3′,

R 5′-GAAATAGGCTTTGTCTTGGTACTG-3′;


*s100a8:*


F 5′-ACAAGGAAATCACCATGCCCT-3′,

R 5′-TGAGATGCCACACCCACTTT-3′;

Relative expression levels were calculated by normalizing levels for genes of interest to that of hprt using the 2–DDCt method.

### 4.9. Statistical Analysis

Statistical Analysis was performed using GraphPadPrism9.4.0 software. Representative data from one of two identical experiments are displayed. The log-rank test for survival curves and Student’s *t*-test or one-way ANOVA with Tukey post-test for multiple comparisons for other tests were used. *p* < 0.05 was considered statistically significant.

## Figures and Tables

**Figure 1 ijms-24-01140-f001:**
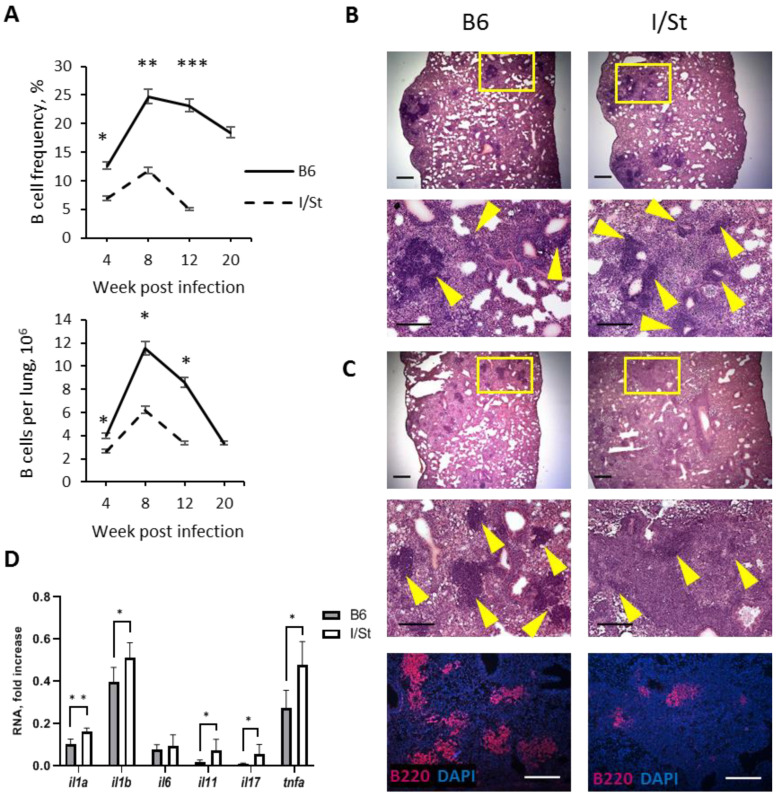
A decrease in number of lung B cells and BCFs coincides with development of diffuse inflammation in TB-susceptible I/St mice. (**A**) Dynamic of relative and absolute B-cell content in the lungs of I/St and B6 mice after *M. tuberculosis* H37Rv infection, FACS enumeration of CD19^+^ cells (percent from all lung cells). (**B**,**C**) Representative pictures of lung pathology in B6 and I/St mice at week 8 (**B**) and 16 (**C**) post-infection. Yellow arrows—B-cell follicles. Scale bar = 500 µm. ((**C**), lower panels) Visualization of B cells with anti-B220 mAbs. Scale bar = 500 µm. (**D**) qrt-PCR evaluation of expression level of genes for inflammatory cytokines as measured in total lung mRNA of B6 and I/St mice at week 16 post-infection. Gene expression was normalized to that of *hprt*. Results of one of two similar experiments are displayed as mean ± SD, 4–5 mice per group per time-point. * *p* < 0.05; ** *p* < 0.01; *** *p* < 0.001.

**Figure 2 ijms-24-01140-f002:**
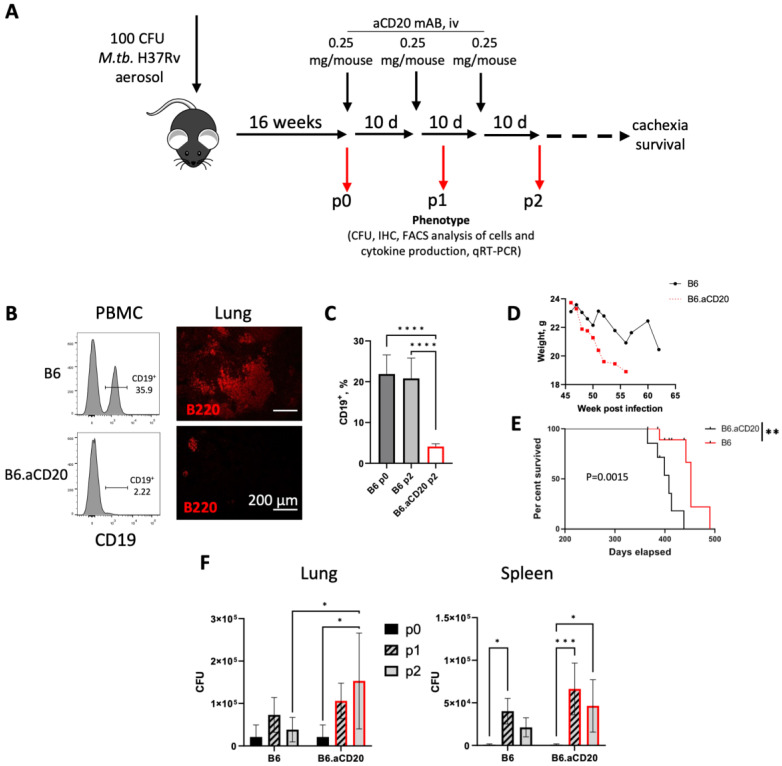
B-cell depletion exacerbates severity of TB at the late stages of infection. (**A**) The scheme of experiments. 0.25 mg of B-cell-depleting anti-CD20 mAbs per mouse per time point were injected intravenously (i.v.) thrice with 10-d intervals. Analyses were performed before treatment (p0) and then at p1 and p2 with 2 wk intervals. (**B**) Efficiency of B-cell depletion as measured by FACS of CD19^+^ B cells in peripheral blood 2 days after antibody injection (PBMC, left panels); visualization of lung BCFs with anti-B220 antibodies 7 days post-depletion (right panels). (**C**) CD19+ B-cell content in the lungs before treatment (p0) and 30 days post-B-cell depletion (p2). Percent from all lung cells. (**D**) Cachexia in control (B6) and anti-CD20 treated (B6.aCD20) mice. (**E**) Lifespan of control (B6) and aCD20 treated (B6.aCD20) mice. Log-rank test, ** *p* = 0.0015. (**F**) Mycobacterial counts in lungs and spleens of control B6 and B-cell-depleted (hereafter—red bars) mice at p0, p1, and p2. Data from one of two similar experiments are displayed as mean ± SD, 4–5 mice per group per time-point. * *p* < 0.05; ** *p* < 0.01; *** *p* < 0.001, **** *p* < 0.0001.

**Figure 3 ijms-24-01140-f003:**
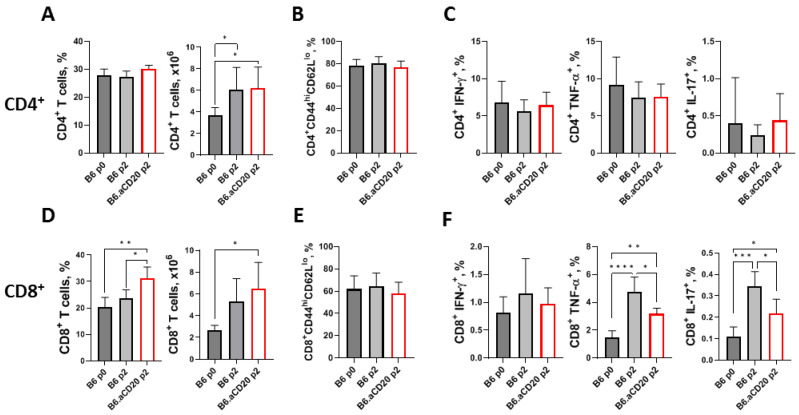
B-cell depletion alters the influx and cytokine production by CD8^+^ but not CD4^+^ T cells. Frequencies and numbers of CD4^+^ (**A**) and CD8^+^ (**D**) T cells in the lungs of B-cell-depleted and control mice. Frequency of activated CD4^+^CD44^hi^CD62L^lo^ (**B**) and CD8^+^CD44^hi^CD62L^lo^ (**E**) T cells in the lungs before and 30 days after anti-CD20 treatment measured by flow cytometry. Intracellular IFN-γ, TNF-α, and IL-17 staining of lung CD4^+^ (**C**) and CD8^+^ (**F**) T cells from control B6 and anti-CD20-treated mice at points p0 and p2 after stimulation in culture with mycobacterial antigens. Data from one of two similar experiments are displayed as the mean ± SD, 4–5 mice per group per time-point. * *p* < 0.05; ** *p* < 0.01; *** *p* < 0.001; **** *p* < 0.0001.

**Figure 4 ijms-24-01140-f004:**
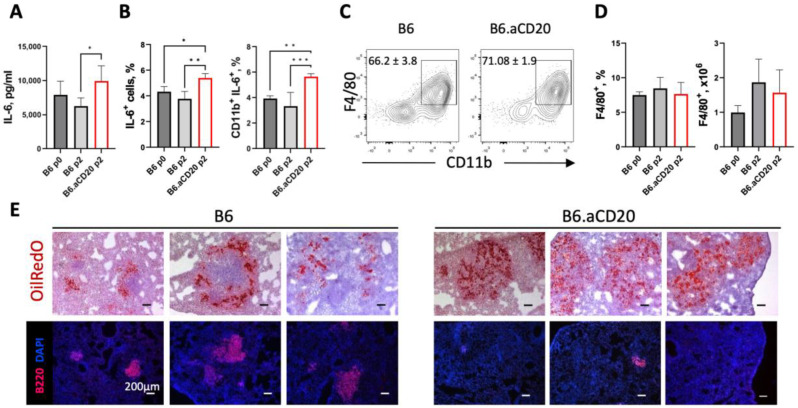
B-cell depletion increases IL-6 production by lung macrophages and elevates the lung content of fat-containing macrophages. Compared to control animals, lung cells from B-cell-depleted mice secrete more IL-6, as detected in culture supernatants by ELISA, (**A**) and display a higher content of IL-6-positive cells (**B**). This population comprises predominantly CD11b^+^F4/80^+^ macrophages (as assessed by FACS for gated IL-6-positive cells, mean ± SD for five individually assessed mice is displayed) (**C**), whose overall numbers and frequencies are similar in control and experimental groups (**D**). Data from one of two similar experiments are presented as the mean ± SD, 4–5 mice per group per time-point. * *p* < 0.05; ** *p* < 0.01; *** *p* < 0.001. (**E**) Visualization of fat-rich macrophages (OilRedO (ORO), red, upper panels) and B cells (lower panels) in lung cryosections from individual B6 and B-cell-depleted mice. Representative areas with varying architecture of macrophage foci and BCFs are displayed. The mean square ± SD of ORO-positive zones in control and B-cell-depleted mice were 7.5 ± 1.7 and 37.5 ± 4.9, respectively, *p* = 0.0079, Student’s *t*-test.

**Figure 5 ijms-24-01140-f005:**
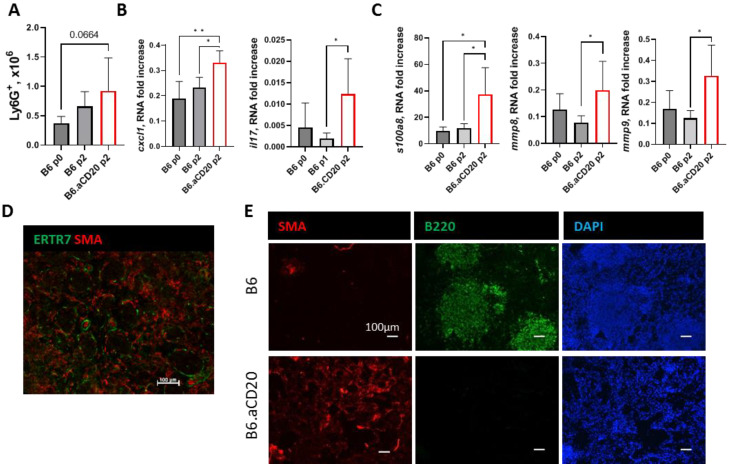
B-cell depletion leads to an increase in expression of genes for neutrophil-associated factors and promotes fibrosis in the lung tissue. (**A**) The content of Ly6G^+^ neutrophils in the lungs of control and B-cell-depleted mice. (**B**,**C**) Expression of genes for neutrophil-recruiting factors CXCL1 and IL-17 (**B**) and neutrophil-produced pathology-associated factors MMP8, MMP9, and S100A8 in the lungs. Gene expression was normalized to that of *hprt*. Data from one of two similar experiments are displayed as mean ± SD, 4–5 mice per group per time-point. * *p* < 0.05; ** *p* < 0.01. (**D**) Fibrosis development in the lungs of TB-susceptible I/St mice at week 9 post-infection with *M. tuberculosis* H37Rv as assessed by staining with anti-SMA (red) and anti-ER-TR7 (green). (**E**) Non-fibrotic B-cell-rich areas (green) in control B6 animals (upper panels) and fibrotic, SMA-positive fibrotic areas (red) in the lungs of B-cell-depleted at the p2 (lower panels). See text for details.

## Data Availability

Not applicable.

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
