# Peer review of "Prolonged B-Lymphocyte-Mediated Immune and Inflammatory Responses to Tuberculosis Infection in the Lungs of TB-Resistant Mice"

_ijms, 2023, doi:10.3390/ijms24021140_

Round 1

Reviewer 1 Report

In this study Linge et al. investigated the role of B cells during late stage TB in mice. After a direct comparison of the presence of B cells and B cell follicles (BCF) in resistant C57BL/6 and susceptible I/St mice they depleted B cells in resistant B6 mice by antibody-mediated depletion. This resulted in a more rapid cachexia and a shortened lifespan of infected animals, which was associated with some immunological changes which I would like to address more specifically in my comments below. Overall the study addresses a timely and relevant question since little is known about the specific role of B cells in late stage TB and many questions regarding their contribution to protection remain unanswered. However, although the reduced survival of B6 mice after late-stage depletion of B cells is an highly interesting and relevant observation, the manuscripts falls short on elucidating the underlying mechanisms. Therefore, before the manuscript can be published I would like to see these comments addressed:

Flow cytometry:

In general, the flow cytometry data description is lacking some detail. It does not become clear how the cells were gated and I would strongly recomment to add the gating strategies for each Figure. Also, for intracellular cytokine staining after ex vivo restimulation, unstimulated controls should  be shown.

·         Figure 1: „FACS enumeration of CD19+ cells“.

Is this % of all cells, of all live cells, of all CD45+ cells?

·         Likewise, Figure 2: „(B) -efficiency of B-cell depletion as measured by FACS of CD19+ B cells in peripheral blood 2 days after antibody injection. (C) CD19+ B-cell content in the lungs before treatment (p0) and 30 days post B-cell depletion (p2).“

Please show gating strategy for the identification of B cells. How did the authors calculate total cell numbers?

·         Figure 3: Please show gating strategies.

·         Figure 4: Please rename (it says Figure 1 again).

Also here the gating is not clear.

·         Line 192: „the vast majority of IL-6 producing cells (more than 60%) were represented by F4/80+ macrophages (Figure 4C).“

Fig. 4C only one representative FACS plot is shown but where is  the quantification of this result for all mice that have been analysed? Please add. Otherwise the statement is not reflected by the data.

Neutrophils:

The data presented do not make a case for neutrophil-associated inflammation. The genes analysed such as S100A8, and matrix metalloproteinases MMP8, MMP9 are not strictly neutrophil-specific. Together with the flow data which do not indicate increased frequencies of neutrophils, the quesion is if increased pathology is indeed neutrophil driven or rather by other cells such as inflammatory monocytes/macrophages. The activation status of these cell poplations should be analysed by flow cytometry to shed light on the pathological processes in the absence of B cells.

Histology:

I am not convinced by th data that are supposed to show fibrosis in the lungs of B cell depleted mice (Fig. 5 D and E). I would rather like to see a Trichrome staining to support the conclusion in lines 240-242:

„massive fibrosis development was evident in the lungs of B-cell depleted animals (Figure 5E), demonstrating for the first time that B-cells play an important role in this aspect of TB pathology.“

Minor comment: Size bar missing in micrographs, please add.

Author Response

  We are grateful for critical comments and suggestions made by all reviewers, many of which helped improving presentation of our results. Below, point-by-point responses to their concerns are provided.

Reviewer 1

Gating strategies for all cell population were added and displayed in Figure S2.   

FACS plots for intracellular staining of Ag-non-stimulated controls are displayed as Figure S3.   

In the Figure 1 and 2 per cent of CD19+ cells of all lung cells is depicted. Corresponding explanations were added to figure legends.

  The number of Figure 4 was changed, a statement that enumeration of F4/80+ macrophages is displayed as mean ± SD for 5 individual mice is added to the legend.

        In a separate study specifically devoted to the role of S100F8/9 protein in a mouse TB model (in preparation, to be published separately) we performed intracellular staining for S100A8/9 dimer for lymphoid and lung cells of infected animals. It appeared that in spleens and lymph nodes this molecule is present exclusively in neutrophils (98-99 per cent of positive cells). In infected lungs, about 11-12% S100A8/9+ cells were macrophages and 88% - neutrophils, which allows considering the latter population as a key one. We prefer not to include this analysis in the present paper since it is just a small part of a big genetic study. Regarding mmp8 (neutrophil collagenase), there is ample evidence that this is predominantly neutrophil enzyme sequestered in granules at the protein level. RNA, of course, does not provide direct prove, but in conjunction with other four genes, two of which encode neutrophil attractants, we feel that the results are highly suggestive.

      Unfortunately, we cannot perform Trichrome staining immediately – we have neither reagents, nor infected mice at hand, let alone B-cell-depleted mice at advanced stages of infection. Moreover, the Editors limited time of revision to 10 days only. All we can do is to make our statement less categorical – the text was changed accordingly. It is also worth mentioning that, upon activation, fibroblasts trans-differentiate into SMA-expressing myofibroblasts demonstrating excessive collagen deposition and tissue remodeling (Pardo A, et al., 2016) and promote stiffness of extracellular matrix (Lomas NJ, et al., 2012). Moreover, it was reported about development of anti-αSMA serological test as a biomarker of activating fibroblasts in lung disorders (Holm Nielsen S, et al., 2019, doi: 10.1016/j.tranon.2018.11.004.).

Bars have been added to micrographs.

Reviewer 2 Report

The results shown in the present study are very well demonstrated and bring good prospects for understanding the immune and inflammatory responses to tuberculosis.

I have few comments to make and modifications to request.

1) The caption of Figure 7 is informing that it is figure 1.

2) Include a graphical quantification of the ORO labeling shown in Figure 7E.

3) On line 415, the citation of a previous work was incorrectly done.

4) Has the use of mice been approved by an ethics committee? Important to mention in the methodology.

Author Response

We are grateful for critical comments and suggestions made by all reviewers, many of which helped improving presentation of our results. Below, point-by-point responses to their concerns are provided.

Reviewer 2

Incorrect number of Fig. 4 was corrected.

To avoid graphical overwhelm, statistics for ORO staining was added to the legend of Fig. 4.

Citation of our previous work looks appropriate.

The manuscript submitted originally contains the Ethical Statement in the first paragraph of Materials and Methods section.

Reviewer 3 Report

The authors reported in this article the importance of B-cells towards the protective nature against TB. The data showed very consistently that B-cells have certain levels of protection and persist for a period of time. There are a few questions that I would like to ask the authors.
1. In Figure 2 E., The Y-axis was named as probability of survival. Would it be more accurate to just put survival?
2. In the same figure, survival of mice showed only a marked improvement of less than 100 days compared to those with depletion of B-cells with CD20. What is the key reason to those without depletion that could only have a slight improvement over the depleted ones.
3. Were there any measure of Tregs in any of the experiments as TB has often been shown to have induce Tregs especially in persisting TB. This is relatively crucial to answer Question 2 above.
4. The reduction of IFN-gamma, TNF-alpha which are key cytokines were also shown in Figure 3 especially with the depletion of B-cells. Why would it be possible that these CD8 cells are not as functional as the non-depleted ones?

Thank you

Author Response

We are grateful for critical comments and suggestions made by all reviewers, many of which helped improving presentation of our results. Below, point-by-point responses to their concerns are provided.

Reviewer 3.

     Thank you for noticing. It was a strange mistake: we always mark survival curves as “per cent survived”, this was corrected.

     Given that the life span of inbred mice normally, in the absence of infection oscillates around 30 months, 3-mo difference looks not a marginal interval, which is confirmed by the P value. And anyway, B-cells are hardly the key defensive factor in TB, we simply know not enough about the subject and try to figure out how different immune responses work during advanced TB.

      We are studying Treg cells in our mouse TB models. Some data were published (e.g., Logunova et al., Proc Natl Acad Sci U S A. 2020; 117(24):13659-13669. doi: 10.1073/pnas.2003170117), some reviewed (Logunova et al., Regulatory T-Cells: Mechanisms of Immune Response Inhibtion and Involvement in the Control of Tuberculosis Infection. Insights Immunol, 2016). We actually estimated the quantity of CD4+FoxP3+ T-cells in the lungs of B6 and anti-CD20-treated animals at p2. It did not differ between the groups.

         No differences for IFN-γ-positive T-cells in groups of mice are displayed in Fig. 3. For TNF and IL-17, we actually see that during observation period the sizes of corresponding CD8 T-cell populations enlarge stronger in control, not in B-cell-depleted mice. Do B-cell serve as attractors of corresponding T-cells, or these cells proliferate in situ better in the presence of B-cells (we think this is more likely) remain to be determined.

Round 2

Reviewer 1 Report

The authors have addressed all my comments I made regarding the first version of the manuscript.